

# On the mechanism of automated fizzy extraction

Chun-Ming Chang[1,2], Hao-Chun Yang[1,2] and Pawel L. Urban[1,3]

[1] Department of Chemistry, National Tsing Hua University, Hsinchu, Taiwan
[2] Department of Applied Chemistry, National Chiao Tung University, Hsinchu, Taiwan
[3] Frontier Research Center on Fundamental and Applied Sciences of Matters, National Tsing Hua University, Hsinchu, Taiwan

Corresponding author
Pawel L. Urban,
urban@mx.nthu.edu.tw

## ABSTRACT

Fizzy extraction (FE) facilitates analysis of volatile solutes by promoting their transfer from the liquid to the gas phase. A carrier gas is dissolved in the sample under moderate pressure ($\Delta p \approx 150$ kPa), followed by an abrupt decompression, what leads to effervescence. The released gaseous analytes are directed to an on-line detector due to a small pressure difference. FE is advantageous in chemical analysis because the volatile species are released in a short time interval, allowing for pulsed injection, and leading to high signal-to-noise ratios. To shed light on the mechanism of FE, we have investigated various factors that could potentially contribute to the extraction efficiency, including: instrument-related factors, method-related factors, sample-related factors, and analyte-related factors. In particular, we have evaluated the properties of volatile solutes, which make them amenable to FE. The results suggest that the organic solutes may diffuse to the bubble lumen, especially in the presence of salt. The high signal intensities in FE coupled with mass spectrometry are partly due to the high sample introduction rate (upon decompression) to a mass-sensitive detector. However, the analytes with different properties (molecular weight, polarity) reveal distinct temporal profiles, pointing to the effect of bubble exposure to the sample matrix. A sufficient extraction time (~12 s) is required to extract less volatile solutes. The results presented in this report can help analysts to predict the occurrence of matrix effects when analyzing real samples. They also provide a basis for increasing extraction efficiency to detect low-abundance analytes.

## INTRODUCTION

Sample preparation—whether performed in manual or automated manner—is frequently an unavoidable step in chemical analysis workflows (*Prabhu & Urban, 2017*; *Alexovič et al., 2018*; *Poole, 2020*; *Zheng, 2020*). It can rely on analyte transfer between different phases in liquid-liquid, solid-liquid, liquid-gas, or solid-gas extraction systems. One of the available approaches is the recently introduced fizzy extraction (FE) approach, which relies on dissolution of a carrier gas in liquid sample under slightly elevated pressure, followed by a sudden decompression of the sample headspace leading to effervescence (*Chang & Urban, 2016*; *Yang, Chang & Urban, 2017*). Although the pressure applied to the sample is

higher than the atmospheric pressure ($\Delta p \approx 150$ kPa), it is still very low in comparison with the pressures utilized in supercritical fluid methods (~8–61 MPa) (*Hawthorne, 1990*). On decompression, multiple (micro)bubbles gush toward the sample surface bringing analyte molecules to the gas phase. The effervescence in FE resembles the phenomenon occurring in shaken soda bottle or in blood vessels of an individual suffering from caisson disease. The surge of analyte molecules in the headspace (on the onset of effervescence) gives rise to high transient analyte signals. FE may be regarded as a pressure-controlled effervescence-assisted emulsification liquid-gas extraction approach.

Fizzy extraction was originally coupled with atmospheric pressure chemical ionization (APCI) mass spectrometry (MS), which enabled real-time monitoring of the released medium-polarity volatile and semivolatile compounds (*Chang & Urban, 2016*). Chemical analysis by FE hyphenated with APCI-MS provides satisfactory detectability because the volatile organic compounds (VOCs) are liberated and reach the ion source of mass spectrometer in a short period of time (few seconds). The entire workflow is expeditious (<5 min). Recently, FE was also hyphenated with gas chromatography (*Yang & Urban, 2019*), and the extraction procedure was supplemented with automated features (*Yang, Chang & Urban, 2019*). Partial automation is critical for repetitive control of the saturation and effervescence steps. Although the FE system has not yet been commercialized, it can readily be built by chemists using inexpensive modules (*cf. Urban, 2018*).

The earlier work on FE raised questions about the possible mechanism responsible for the release of molecules into the gas phase. With the lack of a plausible description of the underlying principles, it is unclear in which cases the procedure can be applied and how to boost its performance. Therefore, in the present study, we attacked the problem of FE mechanism. For that purpose, we first identified a number of factors that can potentially influence FE performance.

## MATERIALS AND METHODS

### Materials and samples

Ethanol (anhydrous, 99.5+%) was from Echo Chemical (Miaoli, Taiwan). Ethyl propionate (EPR), ethyl pentanoate (EPE), ethyl hexanoate, ethyl nonanoate (EN), ethyl undecanoate (EUD), and polyethylene glycol (PEG) 400 were from Alfa Aesar (Ward Hill, MA, USA). Ethyl butyrate (EB) was from Acros Organics (Pittsburgh, PA, USA). Ethyl heptanoate (EHP) was from TCI (Tokyo, Japan). Ethyl octanoate, ethyl decanoate (ED), gum arabic (GA, from acacia tree), and (R)-(+)-limonene (LIM) were from Sigma-Aldrich/Merck (St. Louis, MO, USA). 2-Propanol (Emsure ACS, ISO, Reag. Ph Eur grade), methanol (LC-MS-grade), and water (LC-MS-grade) were from Merck (Darmstadt, Germany). Sodium chloride was from Showa Chemical (Tokyo, Japan). Sodium dodecyl sulfate (SDS; >90%) was from Aencore Chemical (Melbourne, Australia).

Typically, the stock solutions of chemical standards were prepared in pure alcohols (methanol, ethanol, and isopropanol). The stock solution concentration (EPR, EPE, EHP, EN, EUD, and LIM) was 1 M. The test samples containing ethyl esters with various carbon chains and LIM were prepared in alcohol/water mixture at varied percentage

(depending on the experiments). The final concentration of chemical standards in the samples used in most experiments was $5 \times 10^{-6}$ M.

## Apparatus

The extraction process is facilitated by two microcontrollers (chipKIT Uno32 and Arduino Mega 2560), which are programed in C++. The program-controlled pressure changes lead to repeatable bubbling in the extraction chamber. The system also features a number of functions, which reduce human effort. Some of these functions are utilized in the present study (controlling valves and motor at defined time points, displaying experimental conditions on an LCD screen, and triggering MS data acquisition). V1 (solenoid valve) is used to control the flow of the carrier gas (typically, carbon dioxide), and it is connected to the sample chamber (20 mL screw top headspace glass vial with septum cap; cat. no. 20-HSVST201-CP; Thermo Fisher Scientific, Waltham, MA, USA) via a 20-cm section of polytetrafluoroethylene (PTFE) tubing (I.D. = 0.8 mm, O.D. = 1.6 mm, cat. no. 58700-U, Supelco; Sigma-Aldrich, St. Louis, MO, USA). A 14-cm section of soft rubber-like tubing (e.g., silicone tubing, I.D. = 0.9 mm, O.D. = 2.0 mm) is attached to V2 (pinch valve), and it is connected in between a 2-cm section of PTFE tubing (I.D. = 0.3 mm, O.D. = 1.6 mm, cat. no. 58702, Supelco; Sigma-Aldrich, St. Louis, MO, USA) and a 60-cm section of ethylene tetrafluoroethylene (ETFE) extract transfer tubing (I.D. = 1.0 mm, O.D. = 1.6 mm, part no. 1517L; IDEX Health & Science, Lake Forest, IL, USA). During the optimization of extract transfer tubing diameter, various PTFE tubings (I.D. = 0.3 mm, O.D. = 1.6 mm, cat. no. 58702; I.D. = 0.6 mm, O.D. = 1.6 mm, cat. no. 58701; I.D. = 0.8 mm, O.D. = 1.6 mm, cat. no. 58700-U; Supelco; Sigma-Aldrich as well as I.D. = 1.0 mm, O.D. = 1.5 mm, from an unspecified supplier) were used instead of the ETFE tubing. The gaseous extract is transferred from the extraction chamber via those tubing sections (PTFE, soft rubber-like, and ETFE) to the ion source of mass spectrometer.

## Procedure

An automated FE system was disclosed earlier (*Yang, Chang & Urban, 2019*), and it was applied in this study following some modifications (e.g., removing T-junction and Valve 3, thermal printer, wireless control, and real-time data processing) (Fig. 1). The FE process consists of three steps (*Chang & Urban, 2016*):

1. Flushing step: Flushing headspace vapors with carrier gas (typically, carbon dioxide) (60 s) (open Valves 1 and 2 (V1 and V2)).
2. Saturation step: Pressurizing carrier gas in the extraction chamber (open V1, and close V2). Stirring is applied in this step to assist the dissolution of the carrier gas in the sample matrix (60 s).
3. Extraction step: Depressurizing the sample chamber and transferring the gaseous extract to APCI-MS. In the *stage* 1 of this step (close V1, and open V2), the dissolved carrier gas is released from the sample matrix, leading to effervescence. The VOCs are extracted during this short period of time (2 s), and transferred to APCI-MS (see Supporting Information text and Table S1). In the *stage* 2 of this step (open V1 and V2), the carrier

gas flow is switched on again to assist in the transfer of the remaining gas-phase analytes present in the sample headspace (28 s).

The default conditions used in most experiments are as follows: sample, $5 \times 10^{-6}$ M analytes dissolved in 5 vol% of alcohol/water mixture; carrier gas, carbon dioxide; extract transfer tubing, I.D. = 1.0 mm; extract transfer tubing length, 60 cm.

## Mass spectrometry

Unless noted otherwise, the sample chamber was coupled via V2 and 60-cm (ETFE) extract transfer tubing with triple quadrupole mass spectrometer (LCMS-8030; Shimadzu, Tokyo, Japan). It was used in conjunction with DUIS ion source (Shimadzu, Tokyo, Japan) operated in the APCI positive-ion mode. The potential applied to the APCI needle was 4.5 kV. Nebulizer gas (nitrogen) flow rate was 2.5 L min$^{-1}$. Drying gas (nitrogen) flow rate was 15 L min$^{-1}$. The temperature of the desolvation line was set to 250 °C, while the temperature of the heated block was set to 300 °C. The data acquisition was performed in multiple reaction monitoring (MRM) mode (see Table S1 for transitions). The pressure of the collision gas (argon) was set to 230 kPa. The dwell time was 100 ms.

## Data treatment

Enhancement factor (EF) is defined as the maximal signal intensity ($I_{\text{max-extraction}}$) in the extraction step divided by the mean signal intensity (mean$_{\text{flushing}}$) during the flushing step (from 0.49 to 1.00 min):

$$\text{EF} = \frac{I_{\text{max-extraction}}}{\text{mean}_{\text{flushing}}}. \tag{1}$$

Signal-to-noise ratio ($S/N$) was calculated using the maximal signal intensity in the extraction step minus the average intensity of the saturation step (local baseline, from 1.35 to 1.85 min) divided by the root mean square of the blank sample in the extraction step (from 2.04 to 2.97 min):

$$S/N = \frac{(I_{\text{max-extraction}} - \text{mean}_{\text{saturation}})}{\text{RMS}_{\text{extraction (blank)}}}. \tag{2}$$

In one part of this study—in order to compensate for differences in ionization efficiencies of the tested analytes—the corrected signal ($S_{\text{corrected}}$) was calculated based on the maximal signal intensity obtained in FE ($I_{\text{max-FE}}$) multiplied by a correction factor (CF):

$$S_{\text{corrected}} = I_{\text{max-FE}} \times \text{CF}. \tag{3}$$

The CF is defined as the average (averaging interval: 1 min) extracted ion currents (EICs) of the target analytes ($I_{\text{target}}$) obtained in direct liquid infusion to APCI-MS (sample flow rate, 40 μL min$^{-1}$; sample, $5 \times 10^{-6}$ M analytes dissolved in 5 vol.% ethanol/water mixture) divided by the average intensity of EPR ($I_{\text{EPR}}$, used as a reference; averaging interval: 1 min):

$$\text{CF} = \frac{I_{\text{target}}}{I_{\text{EPR}}}. \tag{4}$$

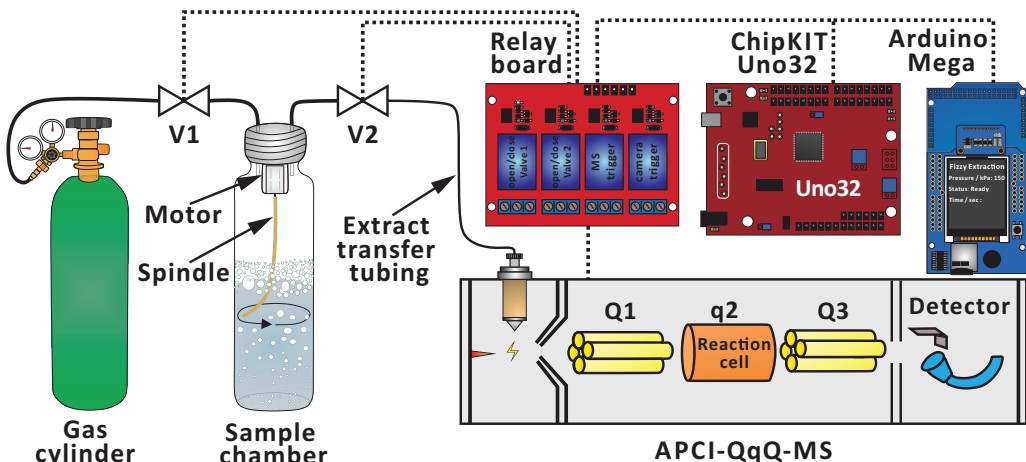

**Figure 1 Simplified scheme of the experimental setup used to conduct FE in conjunction with APCI-MS.**

The calculations were done using Excel (version 16.0; Microsoft, Redmond, WA, USA) and Matlab software (version R2017a; MathWorks, Natick, MA, USA). Origin software (version 2018b; OriginLab, Northampton, MA, USA) was used to plot the figures.

## RESULTS

To shed light on the mechanism of FE, we have studied the influence of a number of parameters on the extraction process. These parameters are grouped into four categories: (1) instrument-related; (2) method-related; (3) sample-related; (4) analyte-related.

### Instrument-related factors affecting FE

#### *Extract transfer tubing diameter and length*

First, we tested 30-cm sections of four types of PTFE tubing with different inner diameters (0.3, 0.6, 0.8, and 1.0 mm) as extract transfer tubing. The EFs and *S/N* increased as the inner diameter increased, especially for EPE, EHP, EN, and LIM (Fig. 2A). More gas-phase analyte molecules could be transferred to the MS ion source per time unit when a tubing with a larger diameter was used. We further verified the influence of the extract transfer tubing length (30, 40, 50, 60 cm; I.D. = 1.0 mm) on the extraction process. The EFs and *S/N* of medium-volatility compounds (EPE, EHP, and LIM) mostly increased with the increasing tubing length (Fig. 2B). This result may be related to the fact that the use of longer tubing leads to a lower gas flow rate (*Coelho & Pinho, 2007*).

### Method-related factors affecting FE

#### *Gas type*

It is known that different gases form bubbles with different size (*Hanafizadeh et al., 2015*). While carbon dioxide is mainly used to produce fizzy drinks, nitrogen is occasionally used (*Hildebrand & Carey, 1969*). Here, five easily available gases (carbon dioxide, nitrogen, air, argon, and helium), with different physical properties, were investigated. Based on the recorded EFs and *S/N*, all the tested gases can be used in FE (Fig. 3A). However, extraction

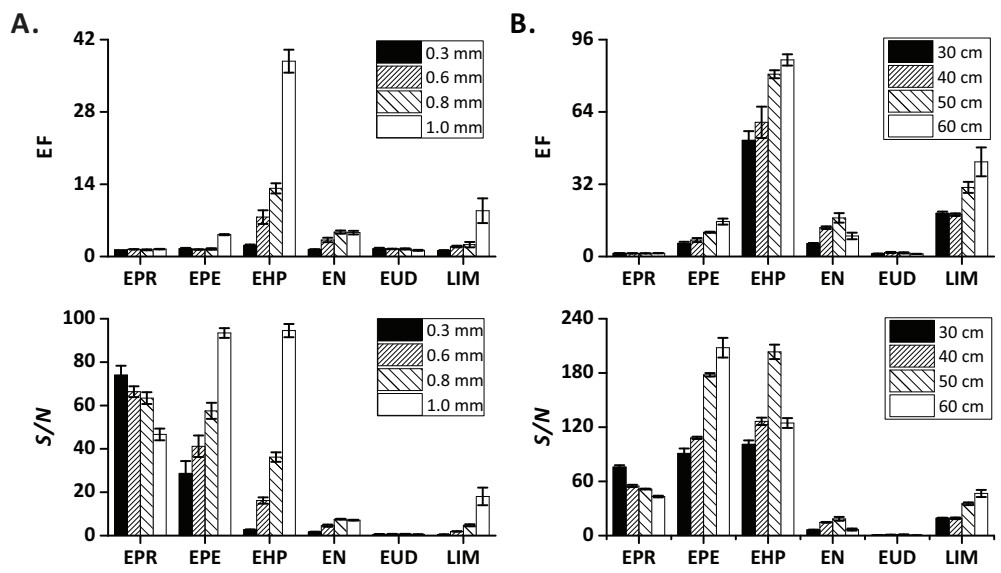

**Figure 2** **Influence of instrument-related factors on EF and S/N.** (A) extract transfer tubing diameter; (B) extract transfer tubing length. Error bars represent standard deviation ($n$ = 3).

of one compound (EPE) was clearly affected by the type of gas used. In the case of carbon dioxide, the EF was the lowest, while in the case of helium, it was the highest. According to Eq. (1), high EF can be either due to high amplitude of signal in extraction step or low amplitude of signal in flushing step. Solubility of gases in water (*Gevantman, 2000*) can influence the dissolution of gases during pressurization, thus affecting the formation of bubbles. Moreover, the densities of the tested gases are dissimilar (*Rathakrishnan, 2004*). These different densities can also contribute to some selectivity in VOC removal from the sample headspace in the flushing step. Owing to the poor solubility of helium in water, and its low density, helium may predominantly flow in the upper part of the headspace in the sample chamber, scavenging highly volatile analytes, which easily diffuse to the upper section of the sample chamber. Therefore, the signals of VOCs during headspace flushing are generally low when using helium as carrier gas (Fig. S1).

### Fast decompression vs. slow decompression

The rationale of this test builds on a daily-life observation: When one slightly loosens cap of a carbonated drink bottle, numerous microbubbles or foam are formed. Conversely, when one unscrews the cap rapidly, the emerging bubbles quickly coalesce, leading to a smaller number of large bubbles. Microbubbles have higher surface area-to-volume ratio than large bubbles (*Zimmerman et al., 2008*). Thus, liquid-gas phase equilibrium should be established in microbubbles faster than in large bubbles.

To emulate the process of slow decompression, we pulsed V2 with different open duty cycles while the cycle duration was fixed at 400 ms. At first, we only applied pulsations in the *stage* 1 of the extraction step (for 2 s). For all the tested conditions, the results did not show clear trends (Fig. 3B), and the temporal profiles recorded in different variants of

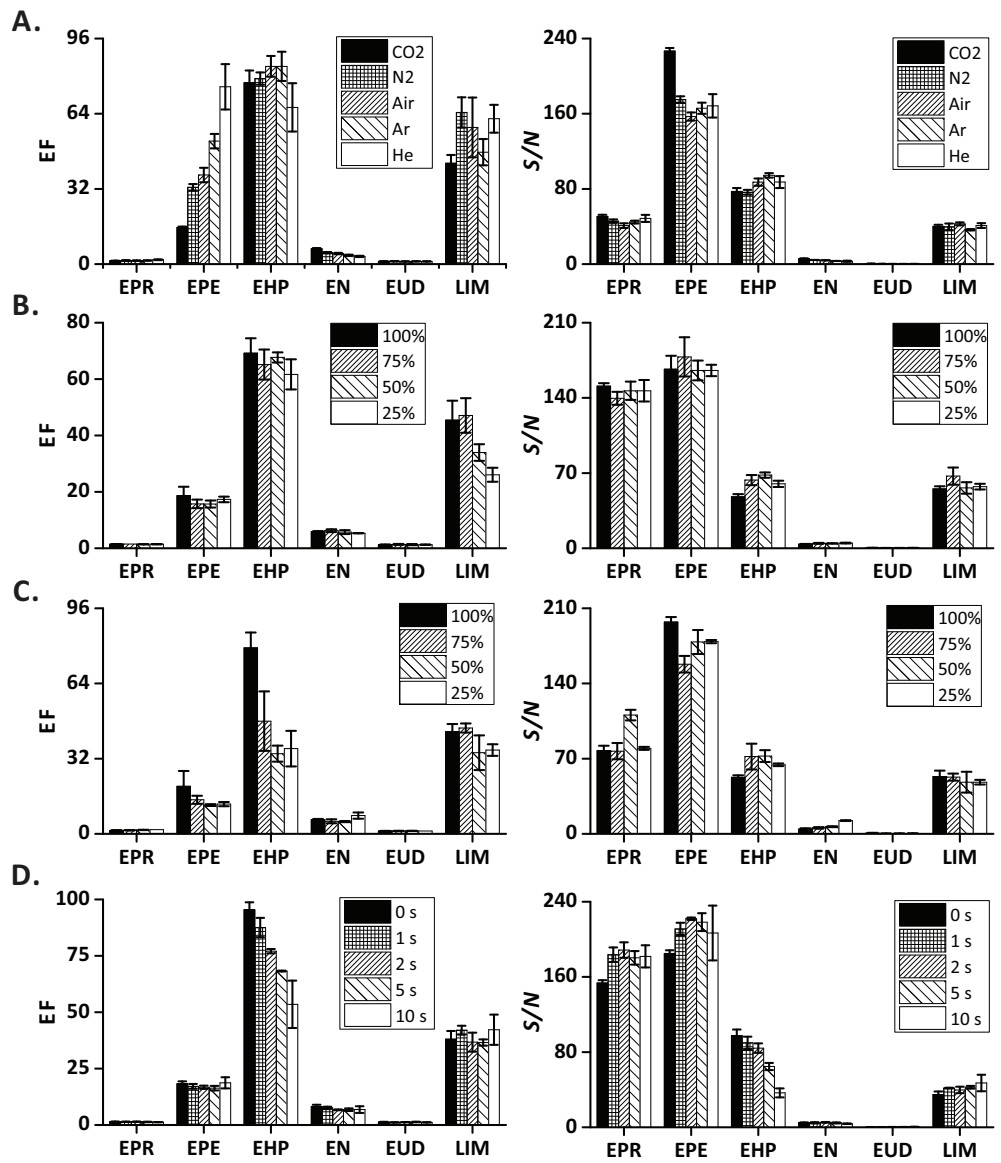

**Figure 3 Influence of method-related factors on EF and S/N** (A) Carrier gas type; (B and C) fast decompression (no pulse was applied) vs. slow decompression (a relay was used to open (o) and close (c) V2 continuously with different "open" duty cycles to provide pulsations; cycle duration, 400 ms; (B) pulsing only in *stage* 1 of the extraction step; (C) pulsing in both stages of the extraction step); (D) extraction time in *stage* 1 of the extraction step (total extraction time, 30 s). Error bars represent standard deviation ($n$ = 3).  

decompression were similar (Fig. S2). In order to verify the effect of slow decompression on FE, we further applied pulsations not only in the *stage* 1 but also in the *stage* 2 of the extraction step (for 30 s). Although no obvious changes were observed, EN exhibited a slightly higher EF and *S/N* with 25% open duty cycle than in other conditions (Fig. 3C). Moreover, the tested analytes revealed distinct temporal profiles when open duty cycles were varied (Fig. S3). The areas under the curve (ion current of the extraction step)

increased with decreasing open duty cycles (Fig. S4). This observation shows that, when the gaseous extract is transferred to MS in pulses (rather than continuously), the sample can still be extracted continuously providing high VOC signals.

We also conducted an experiment, in which we changed the duration of the *stage* 1 of the extraction step. The EFs and *S/N* ratios were not affected to a great extent in most of the tested compounds except EHP (Fig. 3D). The decline in EF of EHP with increasing extraction time may be because the extracted EHP molecules were not concentrated in the headspace of the sample chamber sufficiently, and were transferred to the detector at different time points leading to two temporal peaks (Fig. S5, extraction time, 10 s). The appearance of the first peak is linked to the VOCs that were extracted during effervescence (*stage* 1), while the second peak is linked to the scavenging of the headspace vapors by pumping the carrier gas (*stage* 2). The observation that the second peak was higher than the first peak (EHP and EN, 10 s) could be explained in the following way: The amount of dissolved carbon dioxide was insufficient to extract and scavenge the analytes with higher molecular weights and lower polarities in the *stage* 1. Scavenging of these analytes continued after opening V1.

## Sample-related factors affecting FE
### Sample solvent

Figures 4A–4C show the influence of organic solvents—present in the sample—on the extraction efficiency. Notably, the EFs and *S/N* decreased with increasing ethanol concentration. This effect is rationalized in the following way: Medium and low-polarity VOCs partition to the liquid phase because of their good solubility in ethanol. Moreover, the surface tension of sample matrix decreases with increasing ethanol concentration, which results in larger bubble size (Fig. S6C). Overall, FE can be performed on samples containing common alcohols in a broad concentration range (0.1–25 vol.%). The three tested alcohols provide similar extraction performances. However, ethanol is the solvent of choice due to its lower toxicity. In addition, ethanol is present in alcoholic beverages, which can readily be analyzed by FE. In this case, the sample pretreatment can be reduced to simple dilution with pure water.

### Presence of salt

Addition of an electrolyte into an aqueous solution can affect partitioning of the organic solutes between the liquid phase and gas phase. Here, we investigated the effect of NaCl at varied concentrations on extraction efficiency of the test analytes. Although the EF of the chosen compounds did not reveal major differences, the *S/N* showed clear ascending trends for the increasing concentrations of NaCl, especially for the highly volatile compounds (EPR, EPE, EHP; Fig. 4D). This result can be explained with the occurrence of "salting out effect" (see below) (*Hyde et al., 2017*).

### Presence of surfactants and related additives

It is known that surface-active agents (so-called "surfactants") can drastically influence bubble behavior; reduce rising velocity, prevent coalescence, and deplete/enhance mass transfer (*Takagi & Matsumoto, 2011*). Here, we evaluated the effect of two commonly used

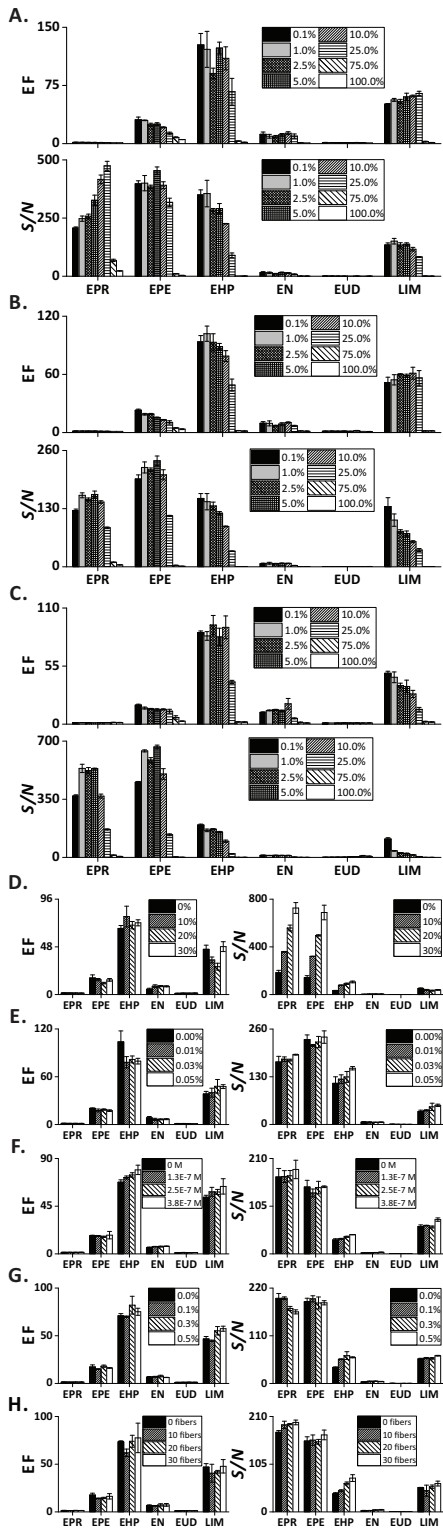

**Figure 4 Influence of sample-related factors (additives in the sample matrix) on EF and *S/N*.**
(A) Methanol; (B) ethanol; (C) isopropanol; (D) sodium chloride; (E) gum arabic; (F) SDS; (G) PEG 400; (H) cotton fibers. In (A–C) and (G) the units are vol.%; in (D and E) the units are wt.%. Error bars represent standard deviation (*n* = 3).

surfactants (GA and SDS)—at concentrations below the critical micelle concentrations (*Moroi, Motomura & Matuura, 1974*)—on the extraction of the target analytes. As expected, the foam height increased notably with the increasing concentration of GA (Fig. S6C). This phenomenon can be explained with the reduction of solution surface tension by GA, which promotes liberation of the dissolved gas (*Cao et al., 2013*), and stabilization of the interfacial film (*Wyasu & Okereke, 2012*). Nonetheless, we only observed a slight increase of EF and *S/N* in LIM (Fig. 4E). Similar to GA, the foaming height increases in the presence of SDS (Fig. S6C), an anionic surfactant. The EF and *S/N* of medium polarity compounds (EHP, EN, LIM) revealed a slightly ascending trend for increasing concentrations of SDS (Fig. 4F).

Because interfacial rheological properties can greatly alter bubble stability by affecting the mass transfer across the interface (*Pelipenko et al., 2012*), we took advantage of the strongly hydrophilic and viscous nature of PEG400 (a surfactant-related additive) (*Harris, 2013*) to evaluate its effect on the extraction performance of the selected analytes. The foaming height of the PEG400-containing sample was greater than the one without adding PEG400 (Fig. S6C). However, only the EF and *S/N* ratios of two compounds (EHP, LIM) were slightly increased when the concentration of PEG400 was increased (Fig. 4G).

### Presence of gas bubble nucleation sites

One technology used by beer industry to control bubbling involves coating the inner walls of bottles with cellulose fibers to increase the abundance of nucleation sites (*Liger-Belair, Polidori & Jeandet, 2008*; *Lee & Devereux, 2011*). Formation of nucleation sites in carbonated liquids with low supersaturation ratios (e.g., carbonated beverages, sparkling wines) requires the presence of gas pockets with radii of curvature larger than the critical nucleation radius (typically, <1 μm) (*Jones, Evans & Galvin, 1999*; *Liger-Belair, 2005*). At this condition, the nucleation energy barrier is overcome, thus promoting bubble growth (type IV non-classical nucleation, according to the nomenclature by *Jones, Evans & Galvin (1999)* and *Lubetkin (2003)*) The tiny and hollow cavities within cellulose fibers (lumens) are responsible for the production of bubble trains, and are regarded as one of the most abundant sources of nucleation sites (*Liger-Belair, Voisin & Jeandet, 2005*). Therefore, we intentionally introduced different numbers of cotton fibers to the sample chamber to increase the abundance of nucleation sites. Although no obvious differences in EFs were observed, a slight increase of *S/N* ratios could be seen (Fig. 4H).

## Analyte-related factors affecting FE
### Analyte properties

To characterize general applicability of FE, we tested a series of ethyl esters with different carbon numbers ($C_5$-$C_{13}$) as model analytes (Table S2). The dependencies of EFs and *S/N* ratios on the investigated physical properties showed distinct trends revealing the amenability of most of the tested compounds to analysis by FE-APCI-QqQ-MS (Fig. 5). This result is in line with the result obtained by FE-GC-Q-MS (*Yang & Urban, 2019*).

In the case of the highly volatile compounds (e.g., EPR, EB), the intensities of EICs in the flushing and extraction steps are similar, resulting in low EFs. This observation is explained

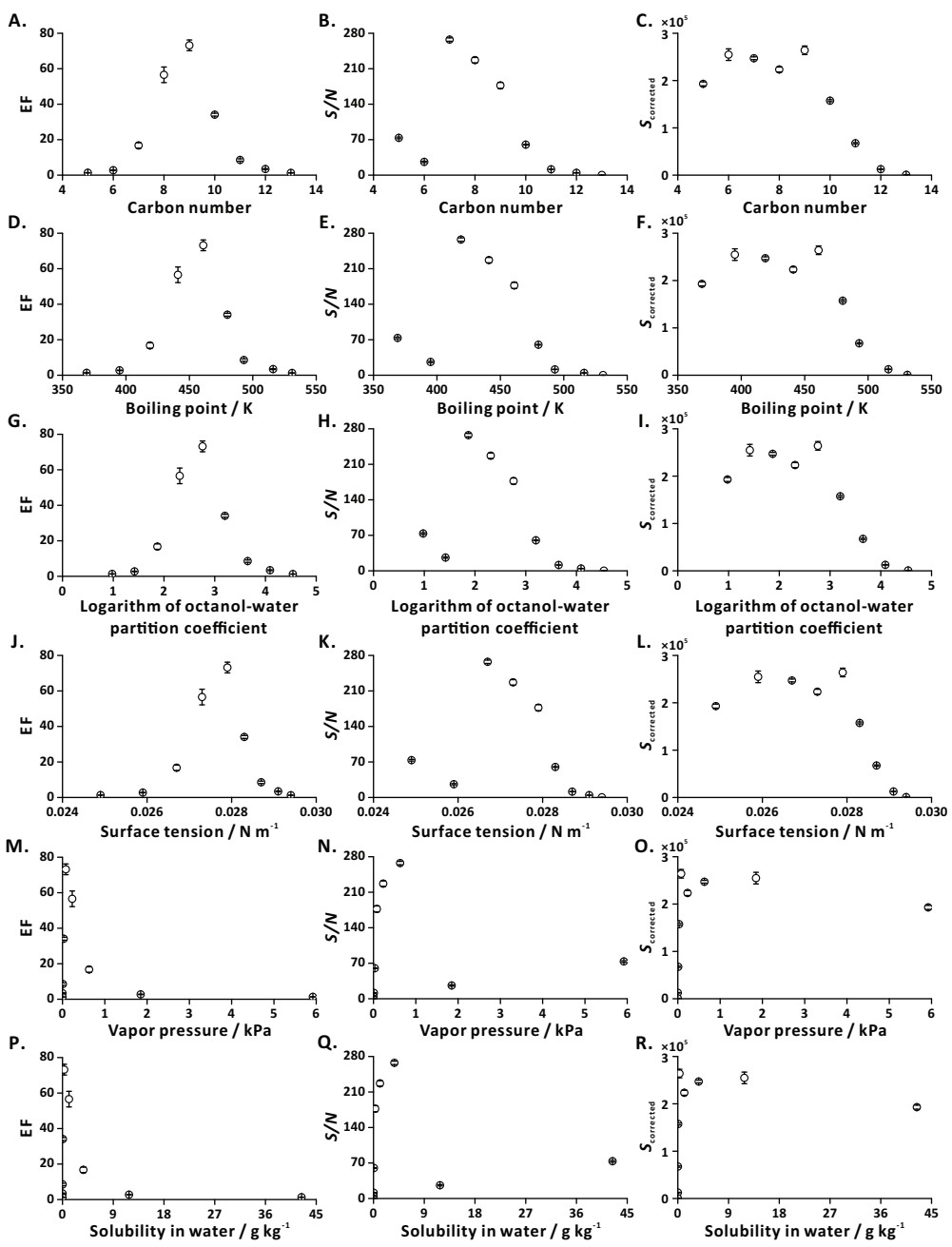

**Figure 5 Influence of analyte-related factors on EF, *S/N* and $S_{corrected}$.** (A–C) Carbon number; (D–F) boiling point; (G–I) logarithm of octanol-water partition coefficient; (J–L) surface tension; (M–O) vapor pressure; (P–R) solubility in water. Error bars represent standard deviation (*n* = 3).

with the fact that the liquid-gas equilibrium is established rapidly. In addition, these polar short-chain ethyl esters strongly interact with solvent molecules, thus limiting their transfer to the gas phase (bubble, headspace) during effervescence. On the other hand, in the case of less volatile compounds (e.g., ED, EUD), the poor EFs may be attributed to their low vapor pressures and high molecular weights. Since the mass transfer of big

molecules into the bubbles (predominantly by diffusion) is limited, only a small portion of these low-volatility species can be transferred to the gas phase, leading to low MS signal intensities.

Ionization efficiencies may differ for the tested analytes, possibly leading to misinterpretation of the results. To compensate for the anticipated ionization bias, we applied CFs (see Eq. (3)) obtained by direct liquid infusion to the FE results. Overall, the analytes characterized with satisfactory EFs, *S/N*, and corrected signal intensities are those that contain 7–10 carbon atoms, have boiling points in the range of 420–480 K, logarithm of octanol-water partition coefficient values in the range of 1.8–3.3, surface tensions in the range of 0.0265–0.0285 N m$^{-1}$, vapor pressure values in the range of 0.03–0.63 kPa, and solubilities in water in the range of 0.08–3.71 g kg$^{-1}$ (Fig. 5). They are highly amenable to FE (EFs ranging from 15 to 75).

### Note on the data variability

To verify the reliability of the results obtained in this study, we further evaluated the reproducibility of the FE technique at the default conditions (carrier gas, carbon dioxide; extract transfer tubing, I.D. = 1.0 mm; extract transfer tubing length, 60 cm; solvent, 5 vol.% ethanol in water). In the case of EFs, the dense distribution of the data points implies good reproducibility among numerous replicates (*n* = 33; 11 days; Fig. S7). However, in the case of *S/N*, the data cloud shows a broader distribution, especially for EPR, EPE, and EHP (Fig. S7). This can be rationalized with the technical variability of the instrument conditions, which may contribute to different MS signal intensities and spectral noise on different days. It is evident that using EFs is a proper way to evaluate extraction performance of FE. By calculating EFs, one can quantify the differences between FE and direct headspace vapor flushing in a single experiment. The use of EFs ensures satisfactory (inter-day) reproducibility, mitigates the effect experimental variability, which would otherwise obscure relevant trends.

### DISCUSSION

It was earlier demonstrated that removal of VOCs from aqueous samples can be greatly enhanced with the aid of microbubbles generated by purging such samples with gases delivered to the bottom of the sample vessel (*Wang & Lenahan, 1984*; *Shimoda et al., 1994*). However, the use of relatively long sample column and large sample volumes are critical to achieve satisfactory extraction efficiencies. To boost analytical performance, the effluent gas extracts are often trapped prior to analysis, as it is in purge-closed-loop and purge-and-trap methods (*Wang & Lenahan, 1984*; *Abeel, Vickers & Decker, 1994*). In general, the gas stripping approach—relying on concentration of the liberated analytes into small volumes—is time-consuming, requires the supply of energy (electricity for heating) and additional consumable materials (sorbent, cryogenic agent).

In FE—unlike in the sparging techniques—numerous microbubbles are formed in situ during fast decompression (Fig. 6). These microbubbles further grow due to absorption of dissolved gas (*Fang et al., 2016*), and coalesce with other (micro)bubbles (*Chaudhari & Hofmann, 1994*). Previously, we showed that increasing carbon dioxide

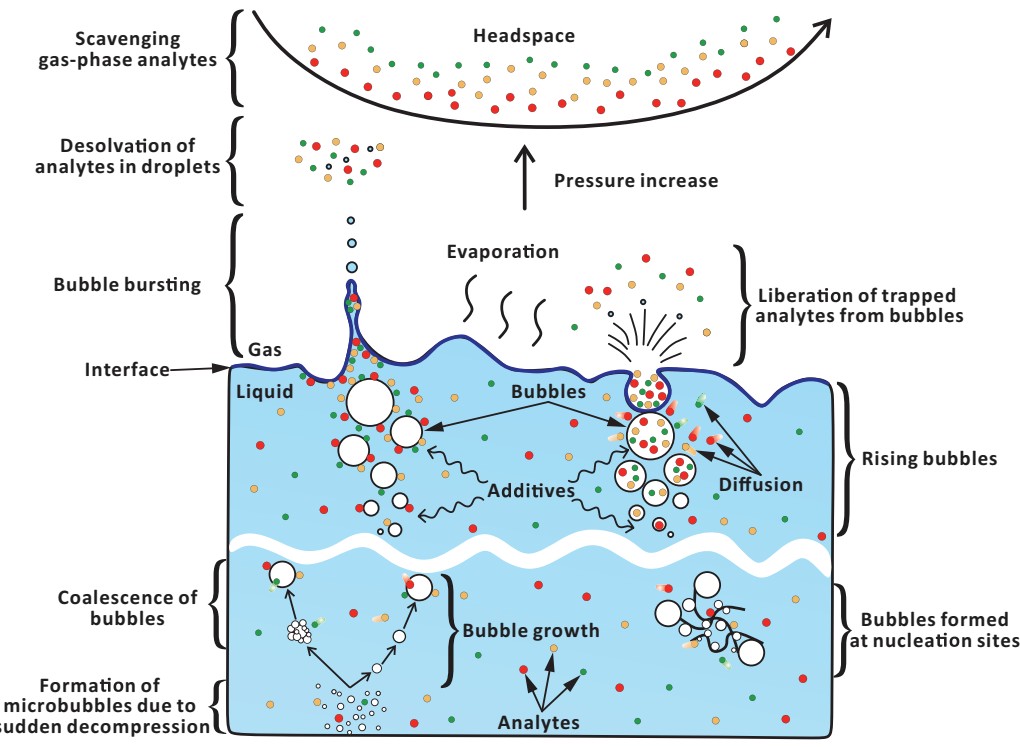

**Figure 6 Hypothetical pathways of analyte transfer from liquid to gas phase in FE.**

pressure (0–150 kPa) during saturation leads to increased analyte peak areas during FE (*Chang & Urban, 2016*). This effect can be attributed to the greater number of microbubbles formed when the concentration of the dissolved gas is higher. However, due to practical and safety-related issues, we did not further utilize higher pressures than 150 kPa. The addition of surfactant can reduce the surface tension, and alter the viscosity of the solution, thus increasing the number of microbubbles. However, the use of surfactants may also cause unmanageable foaming problems leading to contamination of flow lines with sample droplets and foam. In fact, the foaming height increased notably in the presence of surfactant-related additives (Fig. S6C). Thus, in order to prevent contamination of the system, the concentration of surfactants cannot be too high. Only few compounds (EHP, LIM) in some cases showed slightly higher EFs and *S/N* when the surfactant concentration was increased (Figs. 4E–4G). Accordingly, one does not anticipate major matrix effects at low concentrations of surfactants present in the analyzed samples.

Microbubbles can readily form at nucleation sites (e.g., rough surface, lumen of cellulose fibers) (*Jones, Evans & Galvin, 1999*). Thus, in one experiment, we introduced cotton fibers into the sample chamber to provide nucleation sites. The fact that the improvement of FE performance was only minor (Fig. 4H) could be due to an uneven distribution of cotton fibers or insufficient number of microbubbles generated with the aid of the air trapped within fiber lumens. It should be noted that the air entrapment is attributed to the fact that the time required for the liquid to fully invade the lumen by capillary action

is longer than the time required to submerge the fibers (*Liger-Belair, 2012*). However, mechanical stirring—applied in the *stage* 2 of the extraction step—can enhance contact between fiber lumens and solvent, thus eliminating the trapped air and reducing the formation of microbubbles. On the other hand, stirring itself can promote formation of bubbles (*Dean, 1944*).

Fizzy extraction showed poor performance at the high concentrations (>25 vol.%) of alcohols (Figs. 4A–4C). This result is explained with an elevated partitioning of organic solutes to the liquid phase. In addition—according to the Stokes' equation (*Stokes, 1851*)—when the bubble size becomes large, the rise rate of such bubbles increases:

$$R = \frac{\rho g D^2}{18\mu},$$
(5)

where $R$ is bubble rise rate (m s$^{-1}$), $\rho$ is density (kg m$^{-3}$), $D$ is diameter (m), and $\mu$ is dynamic viscosity (kg m$^{-1}$ s$^{-1}$). High rise rates increase the probability of bubble coalescence, thus reducing bubble density (*Rajib, Farzeen & Ali, 2017*). The observed reduction of bubbling and large bubble size at high alcohol concentration (Fig. S6C) can also be attributed to the decreased surface tension (*Rajib, Farzeen & Ali, 2017*). Down this path, few large bubbles provide smaller liquid-gas interface surface area than many microbubbles, while high rise rate shortens the time of their exposure to the sample, what—along the increased partitioning to the liquid phase—leads to low extraction yields.

Enrichment of organic solutes by "bursting bubble aerosols" is based on the assumption that analyte molecules can adsorb on bubble surface during its exposure to sample matrix, and then are released to the headspace via microdroplets ejected from the bubble surface (*Chingin et al., 2016*, *2018*). The number of adsorbed molecules can gradually increase as bubbles grow. Moreover, other solutes can expel some of the analytes from the liquid phase, contributing to adsorption of these analytes onto bubble surface.

The so-called "salting effect" is described by Setschenow equation (*Setschenow, 1889*):

$$\log \frac{S}{S_o} = -K_{salt} C_{salt},$$
(6)

where $S$ is the solubility of the organic solute in aqueous solution, $S_o$ is the solubility of the organic solute in pure water, $K_{salt}$ is the empirical Setschenow constant, and $C_{salt}$ is the molar concentration of the electrolyte. As the concentration of salt increases, the solubility of the organic solutes in the salt solution decreases due to the electrostatic interactions between the salt ions and water dipoles, which reduce the number of freely available water molecules in the solution (*Hyde et al., 2017*). Hydrophobic interactions are promoted leading to aggregation of the organic solutes (*Thomas & Elcock, 2007*). Therefore, the organic species distribute near the liquid-air interface, and are concentrated in the headspace by the bursting bubbles. In the case of highly volatile and polar compounds, a notable increase in S/N ratios but no major differences in EFs were observed in the presence of salt (Fig. 4D). However, no clear trend was observed for the less volatile and less polar compounds. These results suggest that—when a bubble rises up—it preferentially scavenges low-molecular-weight (here, polar) compounds. An adsorption

equilibrium at the bubble surface may be established rapidly when large quantities of low-molecular-weight polar compounds are present near the interface, and only limited space is available for non-polar solutes (with higher molecular weight) to adsorb. Insufficient adsorption space at the bubble surface may be responsible for poor extraction efficiency of less volatile compounds in FE.

The type of carrier gas only had a minor effect on FE. The solubility of the gas greatly influences the amount of bubbles produced during effervescence. Therefore, different MS ion signal intensities are recorded in the extraction step when different gases are used (Fig. S1). Moreover, the five tested gases have non-polar character. Other available gases (e.g., with higher polarity) were not considered mainly due to safety reasons. It is imaginable that compounds with non-polar moieties may adsorb on the bubble surface. However, the signals of low-polarity compounds were lower than those of medium-polarity compounds (Fig. 5H and 5I).

It is also imaginable that the low-molecular-weight (volatile) molecules diffuse into the bubble lumen, or are incorporated into the bubble during bubble growth ("co-precipitation" into the gas phase). Since the applied gases are non-polar, the lower-polarity analytes should, in principle, be preferentially extracted by the bubbles, entering their lumens. However, against intuition, the lowest polarity compounds in the tested set do not show the highest EFs, S/N, and corrected signal intensities (Fig. 5G–5I). Furthermore, the diffusion coefficients of the solutes in the solvent and across the solvent-gas interface can alter their transfer to the bubble surface and across the interface. Note that the diffusion coefficients of gases are related to their molecular weight (*Mason & Kronstadt, 1967*). Turbulence—due to stirring and effervescence—can improve the mass transfer (*Sandoval-Robles, Delmas & Couderc, 1981*) of solutes to the proximity of liquid-gas interface (bubbles, headspace) but it may not necessarily affect the mass transfer across the liquid-gas boundary around the bubbles. When salt is present in the sample, the organic solutes may partition into this boundary. However, the extraction of the lower-polarity analytes was not improved by the addition of salt (Fig. 4D). This may be because of the inefficient mass transfer of big molecules across the liquid-gas interface into the bubble lumens (dominated by diffusion) as well as competition with small molecules (solvents, other analytes). Thus, the efficiency of mass transfer across the liquid-gas interface—rather than chemical similarity (polarity)—may be a more important factor determining extraction rates. Although the low-molecular-weight analytes are highly polar, they more readily "escape" the "crowded" liquid-phase environment (in favor of the "dilute" gas-phase (bubble) environment) than the higher-molecular-weight analytes.

Following exposure of the newly formed bubbles to the sample matrix, the rising bubbles reach the surface of liquid-headspace interface. In the case of jet drops, few (<10) big droplets are formed when the bubble internal cavity collapses (*Resch, Darrozes & Afeti, 1986*). However, in the case of film drops, high numbers (e.g., hundreds, depending on the bubble size) of tiny droplets are produced when the disintegration of the bubble film cap takes place near the liquid-gas interface (*Resch, Darrozes & Afeti, 1986*). It is worth noting that—according to the previous tests (*Chang & Urban, 2016*)—no liquid droplets (which might come from stirred sample and numerous bubbles from

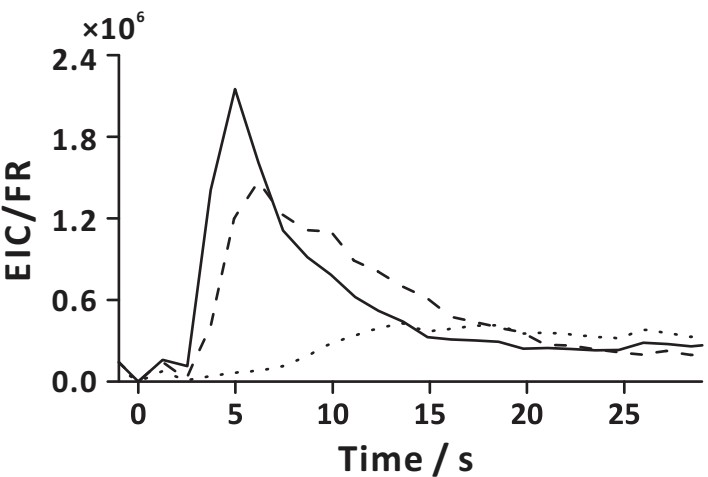

**Figure 7** **Extraction analyte flux as a function of time in the extraction step, defined here as the ratio of extracted ion current (EIC) and momentary carrier gas flow rate (FR).** Solid line represents EPE; dashed line represents EHP; dotted line represents EN. Here, time 0 s corresponds to the start of the extraction step (*stage* 1). The averaged analyte fluxes obtained in three replicate experiments are displayed.

effervescence) enter the detection system. This implies that all the solvent molecules within the droplets evaporate before they reach the detector. Therefore, only gas-phase molecules are ionized and detected by APCI-MS.

The FE technique involves pressurization of carrier gas in the sample chamber, which leads to pressure increase in the headspace. The released gas-phase molecules are directed to the detector due to the pressure difference between the sample chamber headspace and detection system. Increasing extract transfer tubing length (Fig. 2B), and emulating slow decompression by pulsing V2 at a low duty cycle (Fig. S3), reveals the influence of gas flow rate on FE performance. Low flow rate of the gaseous extract extends FE duration and increases the exposure of bubbles to the liquid matrix. The temporal profiles of the carbon dioxide flow rate showed that the gas flow rate increases abruptly when depressurization occurs (V2 open), then it drastically decreases within ~0.7 s after opening V2 (Fig. S8). On turning on the carrier gas supply (V1 open), the gas flow rate again slightly increases, and finally stabilizes (~1.2 s after opening V1). Interestingly, the temporal profile of the initial gas flow rate bears a resemblance to the temporal profiles of the ion currents of some analytes (e.g., EPE, EHP; Figs. S5 and S8, 2 s). It must be pointed out that APCI-MS is a mass-sensitive detector, and the ion signal is proportional to the absolute amount of the analyte molecule entering the detection system in a time unit (sample introduction rate) (*Urban, 2016*; *Prabhu, Witek & Urban, 2019*). Additionally, a sudden release of considerable amount of gaseous effluents into the ion chamber may lead to a transient carryover effect within the ion source. Extended residence of the large amount of gaseous extract in the ion source can lead to peak tailing. Interestingly, different compounds reveal distinct temporal profiles, and their signals reach maxima at different time points (Fig. S5). This may be because the VOCs with various diffusivities are stratified in the headspace, or the fact that it takes more time to extract less

volatile compounds (e.g., EN) by the emerging bubbles (Fig. 7). Nevertheless, with the aid of carrier gas flow (V1 open), larger molecules eventually arrive in the detection system following a short delay (~12 s after opening V2).

## CONCLUSIONS

We have characterized FE process taking into account various factors related to the instrument, method, sample, and analytes. Some of the tested factors (e.g., diameter and length of the extract transfer tubing, alcohol concentrations, and presence of salt) have a significant effect on the extraction performance, while others (e.g., gas type, presence of surfactants and nucleation sites) do not affect it to a great extent. This information provides the basis for boosting extraction efficiency to detect low-abundance analytes. It can also help analysts to predict the occurrence of matrix effects when analyzing real samples. It is proposed that volatile analytes are extracted into bubble lumens. Small molecules more readily diffuse into the bubble lumens, while the diffusion of bigger molecules takes more time. Thus, the analytes with different molecular weights reveal distinct temporal profiles, and are detected at different times. The high amplitudes of the MS signals corresponding to small molecules are attributed to high extraction yields of such species as well as the characteristics of the APCI-MS detection system, which is vulnerable to changes in sample flow rate. Overall, the results delimit the applicability of the FE technique, thus allowing one to predict in which cases the technique can be used, and point out which parameters need to be optimized during FE-based method development.

## ACKNOWLEDGEMENTS

We thank Prof. Chien-Ming Tseng for his help.

### Funding

This work was supported by the Ministry of Science and Technology (MOST), Taiwan (grant numbers 104-2628-M-007-006-MY4, 108-2113-M-007-017, and 108-3017-F-007-003), the National Chiao Tung University, the National Tsing Hua University (grant number 108QI009E1), the Frontier Research Center on Fundamental and Applied Sciences of Matters as well as the Featured Areas Research Center Program within the framework of the Higher Education Sprout Project established by the Ministry of Education (MOE), Taiwan. The funders had no role in study design, data collection and analysis, decision to publish, or preparation of the manuscript.

### Grant Disclosures

The following grant information was disclosed by the authors:
Ministry of Science and Technology (MOST), Taiwan: 104-2628-M-007-006-MY4, 108-2113-M-007-017, and 108-3017-F-007-003.
National Chiao Tung University, the National Tsing Hua University: 108QI009E1.

Frontier Research Center on Fundamental and Applied Sciences of Matters as well as the Featured Areas Research Center Program within the framework of the Higher Education Sprout Project established by the Ministry of Education (MOE), Taiwan.

## Competing Interests

Pawel L. Urban is an Academic Editor for PeerJ.

## Author Contributions

- Chun-Ming Chang conceived and designed the experiments, performed the experiments, analyzed the data, contributed reagents/materials/analysis tools, prepared figures and/or tables, performed the computation work, authored or reviewed drafts of the paper, approved the final draft.
- Hao-Chun Yang conceived and designed the experiments, contributed reagents/materials/analysis tools, performed the computation work, authored or reviewed drafts of the paper, approved the final draft.
- Pawel L. Urban conceived and designed the experiments, contributed reagents/materials/analysis tools, authored or reviewed drafts of the paper, approved the final draft, secured funding.

## Data Availability

The raw data and codes are available at FigShare: Urban, Pawel (2019): Data and code. zip. figshare. Dataset. DOI 10.6084/m9.figshare.10080635.v1.

## Supplemental Information

Supplemental information for this article can be found online at http://dx.doi.org/10.7717/peerj-achem.2#supplemental-information.

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
