# Peer review of "On the mechanism of automated fizzy extraction"

_PeerJ Analytical Chemistry, doi:10.7717/peerj-achem.2_

## Round 0.1 · original submission · Major Revisions

Please proceed with major revision by taking into account all comments and suggestions by both reviewers.

Reviewer 1 ·

Basic reporting

A fizzy extraction (FE) is a novel liquid-gas separation system with emulsification of aqueous sample by effervescence effect. The pressure-controlled changes in the extraction environment heads to the immediate analyte flow through gaseous headspace to the on-line coupled analyser. In this regard, we may consider it as an pressure-controlled effervescence-assisted emulsification liquid-gas extraction approach.

As a very recently developed extraction, it deserves attention to be studied and gradually improved if it would have brought better sensitivity, selectivity, robustness and so forth.

The mechanism of FE and important analytical parameters, have been studied in this work. The instruments, sample, and chemicals were investigated to eliminate potential interfering agents. The chemical conditions were also optimised to get the best mass spectrometry signal.

I miss to stress more on the automation as it gave refinment in repetitive control of the bubbling and effervescence.

(Introduction, Note):
In the very first part of first paragraph, I miss the general starting of how important is the sample preparation when sensitive analysis is to be used. For the record, I would like to give you an example and related citations which significantly contributed in the area. Also there have been important contributions in unattended approaches applied in chemical laboratory. As this field gradually increases in analytical chemistry the also deserve mentioning.

(Introduction, Upgrade):
Sample preparation (whether done in manual or automated manner), which work on the basis of analyte transfer between different phases such as liquid-liquid, solid-liquid, liquid-gas, solid-gas etc., is frequently unavoidable step before analysis [https://www.sciencedirect.com/science/article/pii/S0165993616302916, https://www.sciencedirect.com/science/article/pii/B9780128169117000013, https://www.sciencedirect.com/science/article/pii/S1570023218306214, https://www.sciencedirect.com/science/article/pii/B9780128169063000212]. The one of them is a very recently introduced fizzy extraction (FE) which relies on dissolution of a carrier gas in liquid sample under ...

Experimental design

Again, in this manuscript, would that do not be better to state and discuss more as it is automated method?

For example in title: On the mechanism of automated fizzy extraction?

It is known the extraciton automation has brought many advantages and solely thanks to instrumental performance the analytical parameters improved.
The program controlled pressure changes can head to a repetitive bubbling manner, in your extraction chamber.

Validity of the findings

The mechanism of FE is exhaustively studied with necessary parameters impacting the extraction efficacy.

However, I do not see using any aqueous real sample analysis. I would say, it would be positive to validate the method applicability in doing so.

Additional comments

Authors present a fizzy extraction for the volatile compounds analysis.
The system is modified in comparsion with prior fizzy extractions.
The novelty are clearly declared as detail study of FE mechanism.

Reviewer 2 ·

Basic reporting

The authors of the manuscript "On the mechanism of fizzy extraction" have presented a systematic study on fizzy extraction in order to shed light on its underlying mechanism. Although, fizzy extraction is a variant of conventional "purge and trap", the new technique is lot faster than "purge and trap" and therefore possesses good potential in modern high throughput laboratories. The manuscript is well written and properly presented. The references look okay. The rational provided to validate the hypothesis is scientifically fair.

Experimental design

Experimental design is okay except that the authors missed to evaluate the impact of pressure more systematically on the solubility of carrier gas and the correlation between the soluble carrier gas concentration and the EF values.

Validity of the findings

The presented results are valid. However, the schematic presented in Figure 6 needs additional explanation in the text. It is unclear as to why a polar or hydrophilic compound would move into the bubbles?

Additional comments

Although the authors tried to explain the mechanism of fizzy extraction, some additional factors need to be evaluated such as dependence of EF on the soluble gas concentration. The mechanism or hypothesis as to why a polar compound would move to bubbles needs further explanation.

---

## Round 0.2 · accepted · Accept

I confirm that the revision has been performed according to reviewers' suggestions, therefore it can be recommended for publication.

Sincerely
Victoria Samanidou